# Development and Application of Dual-Polarization Antenna for Dielectric Logging Sensor

**DOI:** 10.3390/s22197667

**Published:** 2022-10-09

**Authors:** Chen Li, Shaogui Deng, Zhiqiang Li, Jian Wu, Baoyin Lu, Jutao Yang

**Affiliations:** 1School of Geosciences, China University of Petroleum (East China), Qingdao 266580, China; 2China Research Institute of Radiowave Propagation, Xinxiang 453000, China

**Keywords:** dielectric constant, slot antenna, dielectric logging sensor, oilfield logging

## Abstract

Because the dielectric constant of water is greater than that of oil and gas, dielectric logging sensors can effectively distinguish oil and gas reservoirs from water layers by measuring the dielectric parameters of formations. Under the special working conditions during the logging of boreholes drilled for oil and gas exploration, such as high temperature and pressure and a narrow working space, the endurance and effectiveness of the antenna in the dielectric logging sensor are crucial. This paper presents a design method for a dual-polarization slot antenna for such working conditions. We theoretically analyzed the working principle of this antenna, established the antenna model, and evaluated its radiation characteristics through simulation. Subsequently, we developed and tested the proposed antenna. The antenna could withstand a high temperature and pressure of 175 °C and 140 MPa, respectively. A dielectric logging sensor using the proposed antenna was successfully applied in oilfield logging.

## 1. Introduction

The oil, gas, and water layers of a formation can be distinguished based on their conductivity and dielectric constant. The conductivity can be measured using a low-frequency electromagnetic wave logging sensor with a multi-turn coil antenna. Based on the variation in the conductivity inside the formation, the oil, gas, and water layers can be differentiated [1,2,3]. However, in the later stages of oilfield development, the fluid distribution inside the formation becomes complex because of freshwater injection for oil displacement, and the oil, gas, and water layers cannot be easily distinguished based solely on conductivity. In recent years, the increased exploration and development of complex formations with shale oil and gas, residual oil and gas, and water-flooded layers have encouraged the development and application of dielectric logging sensors [4,5,6,7].

A dielectric logging sensor uses a small antenna to transmit microwaves to the formation in wells. In Figure 1, the red arrow represents the broadside antenna, and the green arrow represents the end-fire antenna. When the microwaves interact with the formation minerals and stored oil, gas, water, and other fluids, their amplitude undergoes attenuation, and phase offset occurs. For example, the permittivity of water is considerably higher than that of oil and gas, allowing us to distinguish oil and gas from water layers [8,9,10,11,12]. Recently, Baker Hughes launched a commercial multi-frequency dielectric logging sensor [13]. In addition, Schlumberger introduced a multi-frequency dielectric scanner [14], and Halliburton launched an advanced 1 GHz high-frequency dielectric logging sensor [15]. These new commercial dielectric logging sensors operate at a working temperature of 150 °C and high pressure of 100 MPa. However, these companies neither specify the size of antennas nor explain their processing in high-pressure environments. In 2015, Wang Bin and other researchers proposed a V-type broadband antenna for the dielectric logging sensor. However, because of its complex structure and difficulty in realizing high-pressure endurance, the antenna does not meet the actual dielectric logging application requirements [16].

The size of an oil borehole is generally $8\sim 12^{\prime \prime }$, and the depth is generally 3000~5000 m. The maximum working temperature and pressure of the above-mentioned antennas are 150 °C and 100 MPa, respectively, which only meet the basic logging requirements of oilfields. However, with the increasing exploration of unconventional and complex oil and gas reservoirs, the development of small-size dielectric logging sensors with endurance to withstand a temperature and pressure higher than 150 °C and 100 MPa, respectively, is important for the effective evaluation of the characteristics of fluids. 

In this study, we propose a design method and processing technology for a slot antenna. The antenna was theoretically modeled, its working principle was analyzed, and its radiation performance was simulated. Subsequently, a small-size, dual-polarization slot antenna (as shown in Figure 2), which can withstand a maximum temperature of 175 °C and pressure of 140 MPa, was developed. The performance of the antenna was experimentally verified, and the application of the dielectric logging sensor using the proposed antenna was carried out in an actual oilfield.

## 2. Theoretical Analysis

A slot antenna is a slot cut in the wall of an enclosure in the form of a cavity resonator, from which electromagnetic energy is radiated. These types of antennas have a small-sized, light-weight, easy-to-seal, and high-pressure endurance design at a high working frequency (through dielectric loading in the cavity), so they are suitable for the logging of oilfields. The proposed slot antenna (Figure 2) is installed on the metal plate of the dielectric logging sensor. The antenna has a size of $D_{0}\times h_{0}=\emptyset 24\times 30\;\mathrm{mm}$. In Figure 2, the green part is the antenna housing (material 4J29), which ensures the endurance of the dielectric sensor antenna to withstand high voltage and corrosion. The pink cylinder represents two antenna elements (material GH145), perpendicular to each other but not in contact. These elements feed the high-frequency current to the antenna, induce a dual polarization in horizontal and vertical directions, and radiate effective electromagnetic energy to the formation. The yellow part represents the glass (material DM305) with a dielectric constant of approximately 4 filled in the antenna slot [17]. Its function is to ensure the electrical insulation and high-temperature and pressure endurance of the antenna to realize its normal operation under the special working environment of the logging sensor, such as a high voltage of 140 MPa and 175 °C. 

The working wavelength of the antenna in the cavity loaded with dielectric is shown in Equation (1).
(1)$$\lambda =\frac{c}{f\sqrt{\varepsilon_{r}}}=7.4\;\mathrm{cm}$$
where $\varepsilon_{r}$ denotes the dielectric constant of the loaded dielectric in the cavity; λ represents the working wavelength of the antenna in the cavity loaded with dielectric; f is the working frequency value of the antenna 1e9 Hz; c is the velocity value of electromagnetic wave propagation in vacuum 3×10e8 m/s. In order to establish a more accurate model in the process of analyzing the antenna radiation and receiving electromagnetic fields, the antenna should be regarded as a point magnetic dipole source by making the aperture size of the slot antenna considerably smaller than its operating wavelength [18]. Therefore, the antenna slot sizes were designed with a long side value $L_{1}=0.14\lambda =1.1\;\mathrm{cm}$, a wide boundary value $w_{0}=0.1\lambda =0.8\;\mathrm{mm}$, and a height value $h_{3}=0.14\lambda =12\;\mathrm{mm}$. Due to the small size of the antenna, it operated in a non-resonant state; therefore, the input impedance resistance of the antenna was small. In order to achieve excellent electrical contact between the antenna and the feed coaxial line and to improve the resistance of its impedance, three measures were taken. (1) The probe length was increased and designed using coaxial line theory. The impedance calculation formula of the coaxial cable is shown in Equation (2).
(2)$$z_{0}\approx \frac{138}{\sqrt{\varepsilon_{r}}}\log_{10}\frac{D_{2}}{d_{0}}$$
where $\varepsilon_{r}$ denotes the dielectric constant value of the loaded dielectric in the cavity 4.4; $z_{0}$ is the characteristic impedance value of the coaxial cable 50 Ω; $D_{2}$ is the external diameter value of antenna feed ∅3.5 mm; $d_{0}$ is the diameter value of the antenna probe ∅0.6 mm. (2) The light brown area in Figure 2 is the laser welding groove, and its position is selected at heights of $h_{2}=0.27h\approx 3.3\;\mathrm{mm}$, and its diameter is $D_{1}=\emptyset 1\;\mathrm{mm}$. (3) The central axis of the end-fire antenna probe and the broadside antenna probe shall be located in the middle of the long and wide edges of the slot opening surface, respectively. The broadside antenna probe is located at $\frac{w_{0}}{2}=4\;\mathrm{mm}$, the end-fire antenna probe is located at $\frac{L_{1}}{2}=5.5\;\mathrm{mm}$ (Figure 2d,e). Based on the requirements for antenna installation and fixation, other size parameters of the antenna, such as $D_{0},\;h_{0},L_{0},\;L_{2},\;h_{1},\;w_{1},\;D_{3\;}$, were selected, respectively, as ∅24 mm, 30 mm, 3.5 mm, 8 mm, 5 mm, 21.5 mm, and ∅4 mm. 

The proposed antenna structure has several advantages: it is easily fabricated; its performance depends on the size of the current in the loop; and it is relatively unaffected by environmental changes.

### 2.1. Performance Analysis of Single Antenna

The working principle of each antenna is simplified in Figure 3a. The fan wire indicates that the antenna probe is in contact with the inner conductor of the feeding coaxial line. The green part represents the housing of the antenna slot in contact with the outer conductor of the feeding coaxial line. The red solid line arrows represent the high-frequency current fed through the inner conductor of the feeding coaxial line. The current passes through the antenna probe, antenna housing, and outer conductor of the feeding coaxial line and generates electromagnetic radiation [19].

The antenna pin, the extension of the core of the feeding coaxial cable, touches the opposite side of the slot. Because the wavelength of the antenna (λ=7.4 cm) working in a medium is greater than the longer side of the aperture (Δl=1.1 cm), the antenna can be regarded as a magnetic dipole located at the origin (Figure 3b). The solid blue arrow represents the magnetic moment, which is equal to the product of the high-frequency current and the equivalent area of the antenna ($\bar{m}=IS\vec{z_{0}})$.The electric and magnetic fields generated by it are shown in Equations (3)–(5):(3)$$\vec{E_{\emptyset }}=\frac{ISk^{2}}{4\pi r}\sqrt{\frac{\mu }{\varepsilon }}\left(1+\frac{i}{kr}\right)e^{ikr}sin\theta \vec{\emptyset_{0}}\;,$$
(4)$$\vec{H_{r}}=-\frac{ISki}{2\pi r^{2}}\left(1+\frac{i}{kr}\right)cos\theta e^{ikr}\vec{r_{0}}\;,$$
(5)$$\vec{H_{\theta }}=-\frac{ISk^{2}}{4\pi r}\left(1+\frac{i}{kr}-\frac{1}{k^{2}r^{2}}\right)sin\theta e^{ikr}\vec{\theta_{0}}\;.$$

The basic oscillator of the magnetic dipole can be considered equivalent to a magnetic dipole with a distance of Δl and magnetic charges at both ends of $+q_{m}$ and $-q_{m}$, respectively.
(6)$$\vec{m}=q_{m}\vec{\Delta l}=q_{m}\Delta l\vec{z_{0}}=IS\vec{z_{0}\;}\;$$

Further, the magnetic current of the magnetic dipole basic array can be obtained as shown in Equation (7), and the corresponding magnetic current is expressed as Equation (8): (7)$$i_{m}=\frac{dq_{m}}{dt}=\frac{S}{\Delta l}\frac{di}{dt}=\frac{S}{\Delta l}\frac{d\left[J_{m}\cos\left(\omega t+\varphi \right)\right]}{dt}\;,\;$$
(8)$$I_{m}=i\omega J_{m}e^{i\varphi }\frac{S}{\Delta l}=i\omega I\frac{S}{\Delta l}\;,$$
(9)$$IS=\frac{I_{m}\Delta l}{i\omega }\;.$$

After substituting Equation (9) into Equations (3)–(5), Equations (10)–(12) are obtained:(10)$$\vec{E_{\emptyset }}=\frac{I_{m}\Delta lk^{2}}{4\pi r\omega i}\sqrt{\frac{\mu }{\varepsilon }}\left(1+\frac{i}{kr}\right)e^{ikr}sin\theta \vec{\varphi_{0}}\;,$$
(11)$$\vec{H_{r}}=-\frac{I_{m}\Delta lk}{2\pi r^{2}\omega }\left(1+\frac{i}{kr}\right)cos\theta e^{ikr}\vec{r_{0}}$$
(12)$$\vec{H_{\theta }}=-\frac{I_{m}\Delta lk^{2}}{4\pi r\omega i}\left(1+\frac{i}{kr}-\frac{1}{k^{2}r^{2}}\right)sin\theta e^{ikr}\vec{\theta_{0}}.$$
where k represents the electromagnetic propagation constant in the formation; Δl is the opening size of the antenna; ε denotes the formation dielectric constant; μ represents the relative permeability of the formation; I is the high-frequency time–harmonic current; S represents the effective area of the antenna loop current; r is the distance between any point in space and the center of the magnetic dipole; $\vec{E_{\emptyset }},\;\vec{\;H_{r}},\;\vec{H_{\theta }}$ are the electromagnetic field strengths of the antenna in the spherical coordinates: $\vec{E_{\emptyset }}$ is the electric field intensity in the direction of ∅, and $\vec{\;H_{r}},\;\vec{H_{\theta }}$ are the strengths of the magnetic field components in the direction of r and θ, respectively; $\vec{\emptyset_{0}}\;,\;\vec{r_{0}},\;\vec{\;\theta_{0}}$ are the unit vectors in the direction of ∅, r, and θ of the spherical coordinate system, respectively. 

The dielectric logging sensor comprises two transmitting antennas, T1 and T2, in the middle and four receiving array antennas, R1, R2, R3, and R4, placed on both sides (Figure 4d). The red arrow represents the broadside antenna, and the green arrow the end-fire antenna (Figure 2). 

Assuming that the transmitting antenna is located at the origin, the antenna only receives the magnetic field component in the direction of its equivalent magnetic dipole, not the magnetic field component orthogonal to the magnetic dipole. Therefore, the magnetic field component received by the broadside antenna is $H_{\theta }$, and the magnetic field component received by the end-fire antenna is $H_{r}$. According to Equations (9) and (10), if kr is greater than 1, the amplitudes of the signal received by the broadside antenna and end-fire antenna are proportional to 1r and 1r2, respectively. With the same spacing, the signal received by the broadside antenna is stronger than that received by the end-fire antenna. The dashed red shaded area in Figure 4d is formed by the intersection of the power radiation patterns transmitted and received by the broadside antenna, indicating that most of the received signals propagate through the thick mud cake and the shallow layers of the intrusion zone. Further, the dashed green shaded area in Figure 4d is formed by the intersection of the transmitting and receiving patterns of the end-fire antenna, indicating that most of the received signals propagate through the thin mud cake and the deep layers of the intrusion zone. Therefore, the detection depth of the end-fire antenna is greater than that of the broadside antenna. In the antenna radiation coordinate system, the x-axis is the borehole direction, and the z-axis is the stratum. The radiation of the antenna in the xy-plane is an “eight-shaped” pattern (Figure 4a), and radiation patterns in the xz and yz planes point to the formation plane (Figure 4b,c).

The 3D antenna radiation gain diagram (Figure 5) indicates that the radiation intensity in the direction of the formation (*z*-axis) is the strongest. The radiation of the broadside antenna shown in Figure 5a is along its horizontal probe direction (*y*-axis). The radiation of the end-fire antenna shown in Figure 5b is perpendicular to the probe direction (*x*-axis), indicating that the antenna with the working frequency of 1 GHz will have the strongest radiation along the probe direction; this proves that the antenna works as a magnetic dipole. However, the antenna gain is about −43 dB, and the low efficiency is about −49 dB. To improve the radiation efficiency of the antenna, an antenna impedance matching circuit is designed.

Antenna impedance matching circuits come in various forms, such as lumped-element networks and microstrip lines. A simpler impedance matching network is usually cheap and reliable with minimum loss. 

In this study, we preferred a hybrid of microstrip and lumped element networks. The impedance matching of the microstrip line and parallel capacitance type is shown in Figure 6. The impedance of the antenna is represented by point A in the impedance diagram. First, the antenna is connected to a microstrip line, the length of which is designed to make the real part of the antenna admittance 0.02 and reach point B in the impedance diagram. Subsequently, a capacitance is designed to make the imaginary part of the antenna admittance zero and reach point C, corresponding to 50 Ω in the impedance diagram. 

### 2.2. Transmission Performance Simulation of Dual Antennas

Based on the antenna element shown above, we simulated two antenna elements spaced 12 cm apart in water ($\varepsilon_{r}=80$, σ=2 S/m), as shown in Figure 7. Antennas 2 and 4 and 1 and 3 were in broadside and end-fire configurations, respectively.

At a working frequency of 1 GHz, the values of the transmission parameters S31 of the broadside antenna and S42 of the end-fire antenna are about −100 dB, and the cross-polarization isolation is greater than 20 dB (Figure 8). The transmission parameters between the antennas can be improved considerably through impedance matching. The actual measurement data showed that the transmission parameters increased by approximately 30 dB after impedance matching (Figure 9).

## 3. Antenna Development and High-Temperature and Pressure Test

### 3.1. Antenna Development

The slot dual polarized antenna is mainly composed of four parts: antenna housing, an end-fire antenna dipole, a broadside antenna dipole, and loaded dielectric. In Figure 10a, the green part is the antenna housing. The two orthogonal pink curved cylinders are the end-fire antenna dipole and broadside antenna dipole, respectively. In Figure 10b, the top center of the antenna is the loaded dielectric.

Compared to the existing designs, based on the optimal balance between antenna performance and small size, the design of the antenna structure is improved in two aspects. (1) Through the integrated design of the dipole and the feed end of the antenna, the dual polarized array is installed in a narrow space. In this way, the antenna input impedance resistance is increased by increasing the length of the antenna pins. As a result, the antenna and feed coaxial are excellently electrically connected, and the energy transfer is improved, further increasing the antenna radiation efficiency. (2) For a small antenna to work normally in a special logging environment with a temperature of 175 °C and a pressure of 140 MPa, the specialized design process includes material selection, machining, metal heat treatment, probe assembly, laser welding, grouting, the sintering of glass, metal plating, a high-temperature and pressure test, and post-treatment, as shown in Figure 11.

The antenna shell and feed oscillator are made of high-temperature-and-pressure-resistant alloys, GH145 and 4J29, respectively. The expansion coefficients of these alloys are close to those of glass, ensuring a good combination with glass. The antenna feed oscillator with a diameter of 1 mm requires fine machining, and a special lathe is used to achieve the accuracy of a small-size feed array by reducing the linear speed of turning. The metal heat treatment process includes high-temperature wet hydrogen, purification and degassing, and peroxidation treatments to form a dense oxide film on the outer surface of the metal. The oxide film firmly adheres to the metal matrix and molten glass. Antenna probe assembly includes welding and sintering. The welding ensures two things: (1) the two probes of the antenna are relatively vertical without contacting each other, especially the curved part of the arc; (2) the end of the probe is welded to a point closest to the metal shell to avoid poor contact between the antenna vibrator and shell owing to virtual welding. Before sintering is performed, a pre-fabricated glass body is tightly installed into the antenna gap, and the gap is further filled with glass powder to avoid voids in the antenna glass during the sintering process. Laser welding is used to weld the different materials (GH145 and 4J29) of the antenna. During welding, the closed structure at the upper end of the antenna vibrator is designed as an open structure, which solves the difficulty of the narrow welding space in the small hole and ensures position accuracy. The grouting and sintering of glass are divided into two steps: pre-sintering and grouting. First, a square glass block is pre-fired to fill the lower part of the antenna vibrator. Because the density of the pre-fired glass block is very high and the gas generated by the additive is fully discharged, it is easier to control the deformation and sintering quality of the vibrator. Subsequently, a glass tube is filled in the narrow part between the feeding vibrator and shell, and powdered glass is used to grout the joints between the shell and vibrator. Metallic gold plating or gilt is used on the antenna feeder terminal to ensure the good electrical performance of the antenna dipole.

During the development process, the steps in Figure 10 solve the problems that occur owing to the electrical and mechanical properties of the antenna such as the high probability of sintered glass having pores, the easy deformability of the pins of the antenna, the incomplete welding of the contact parts between the pins and shell, and inconsistency in the direction of the vibrator. The performance of the small antenna under a high pressure of 140 MPa and a temperature of 175 °C was verified; the verification device for this purpose is shown in Figure 12a. Figure 13 shows the specially designed borehole, which was the testing environment for the antenna.

### 3.2. Measurement of High-Temperature and Pressure Endurance of Antenna 

A total of four sealing rubber rings were installed on the upper and lower parts of the inner wall of the antenna pressing device to realize sealing and pressure endurance on the bottom and outside of the antenna mounting cavity, respectively. The antenna was fixed at the antenna mounting cavity. The external pressure was only applied through the end face of the antenna (Figure 12).

By simulating the high-temperature-and-pressure working environment of the dielectric sensor in the specially designed test borehole, the three stages of pressure rise, pressure holding, and pressure relief can be observed, especially when the pressure fluctuates greatly when maintaining a certain pressure curve. A fluctuation indicates that the end face of the antenna cannot withstand high pressure, and there is pressure leakage; in which case, the antenna should be taken out and checked for liquid in ports 1 and 2 and at the bottom of the antenna installation cavity. If there is liquid in the ports and cavity, it means the antenna cannot withstand the high temperature of 175 °C and high pressure of 140 MPa.

As shown in Figure 13a, when a pressure of 140 MPa and temperature of 175 °C is maintained in the borehole (as shown in the red square frame), there is no sudden drop in the pressure and temperature curve. Furthermore, there is no fluid leakage in the cavity of the detection antenna slot. The test verified that the antenna line could withstand the high temperature of 175 °C and high pressure of 140 MPa.

The S11 amplitude of the developed antenna varies with temperature, as shown in Figure 14. The S11 amplitude of the antenna with a working frequency of 1 GHz is less than −10 dB in the operating temperature range of 20 °C to 175 °C, which meets the application requirements.

## 4. Practical Logging Application in Oilfield

The Songliao Basin oilfield, located in Northeast China, contains rich unconventional shale oil and gas resources. The dielectric logging sensor using the proposed antenna was successfully applied in the exploration of shale oil reservoirs in the basin. The results of the dielectric logging sensor interpretation of a shale oil borehole are shown in Figure 15. 

The first curve is the borehole depth curve with a depth of 2310–2355 m. In the second channel, GR represents a conventional gamma logging curve, which can be used for lithology identification to provide the shale content of the formation. CAL denotes the borehole diameter, and BS is the diameter of the drilling bit. The third channel shows RT10, RT20, RT30, RT60, and RT90 curves representing the formation resistivity measured using a conventional array induction logging sensor (10–150 kHz). The resistivity of layers 25–30 covers one order of magnitude, ranging from 10 to 20 Ω·m. Therefore, from the conventional resistivity curve measured using a conventional array induction logging sensor, it is difficult to make a distinction between the dry and oil layers. The fourth column shows DEN, the compensated density logging curve; CNL, the compensated neutron logging curve; and AC, the acoustic logging curve, which can provide the total porosity of the formation.

The fifth column shows the HR41 and HR32 curves; these are the resistivity values (in the horizontal polarization direction of the formation) provided by the broadside antennas of the dielectric logging sensor. The sixth column shows the VR41 and VR32 curves; these are the resistivity values (in the horizontal polarization direction of the formation) provided by the end-fire antennas of the dielectric logging sensor. The seventh channel shows the HD41 and HD32 curves; these are the dielectric values in the formation horizontal polarization direction provided by broadside antennas of the dielectric sensor. The eighth channel shows the VD41 and VD32 curves; these are the dielectric values (in the vertical polarization direction of the formation) provided by the end-fire antennas of the dielectric logging sensor. In the ninth channel, PORT is the total formation porosity (red solid line) determined from the intersection of the neutron density in the fourth column; SW and PORW are, respectively, the formation water mineralization and water porosity based on the dielectric constant logging curve. Because the oil-bearing porosity of the formation is the difference between PORT and PORW, the oil-bearing characteristics of the formation can be directly evaluated by comparing the PORT and PORW values in the ninth column. Through a comprehensive analysis, layers 27 and 30 were determined to be class I oil layers; layers 25, 26, and 28 were determined to be class II oil layers; and layer 29 was defined as the dry layer. 

## 5. Conclusions

Based on the theoretical analysis and simulation of a small antenna, we present antenna processing and impedance matching methods suitable for applying dielectric logging sensors in oilfields. We developed a dual-polarization slot antenna to be used in a dielectric logging sensor. The testing established that the proposed antenna can withstand a high temperature and pressure of 175 °C and 140 MPa, respectively. The antenna underwent a practical logging application in an oilfield. The antenna was found to be suitable for dielectric logging and could effectively evaluate the oil-bearing characteristics of the formation. Although these antenna applications have only been tested in shale oil reservoirs in the Songliao Basin, they can provide a basis for finding suitable applications of the antenna in other unconventional reservoirs in domestic oilfields.

## Figures and Tables

**Figure 1 sensors-22-07667-f001:**
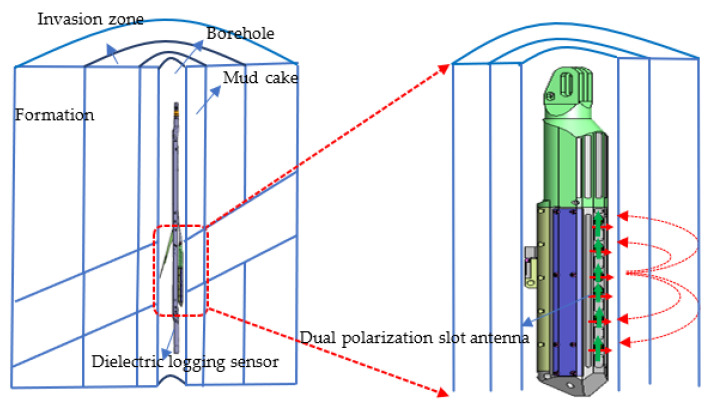
Schematic diagram of dielectric logging sensor.

**Figure 2 sensors-22-07667-f002:**
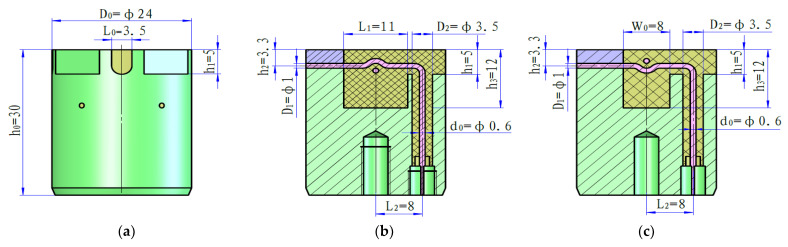
Three-dimensional model and physical dual-polarization slot antenna (mm): (**a**) three-dimensional model; (**b**) frontal anatomical view; (**c**) lateral anatomical view; (**d**) top view; (**e**) physical top view; and (**f**) physical side view of the antenna.

**Figure 3 sensors-22-07667-f003:**
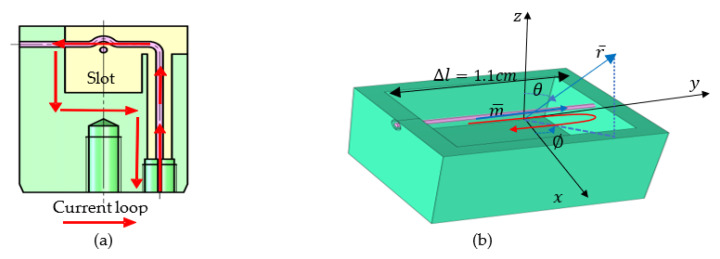
Schematic diagram of slot antenna: (**a**) working principal diagram of antenna; (**b**) antenna equivalent model.

**Figure 4 sensors-22-07667-f004:**
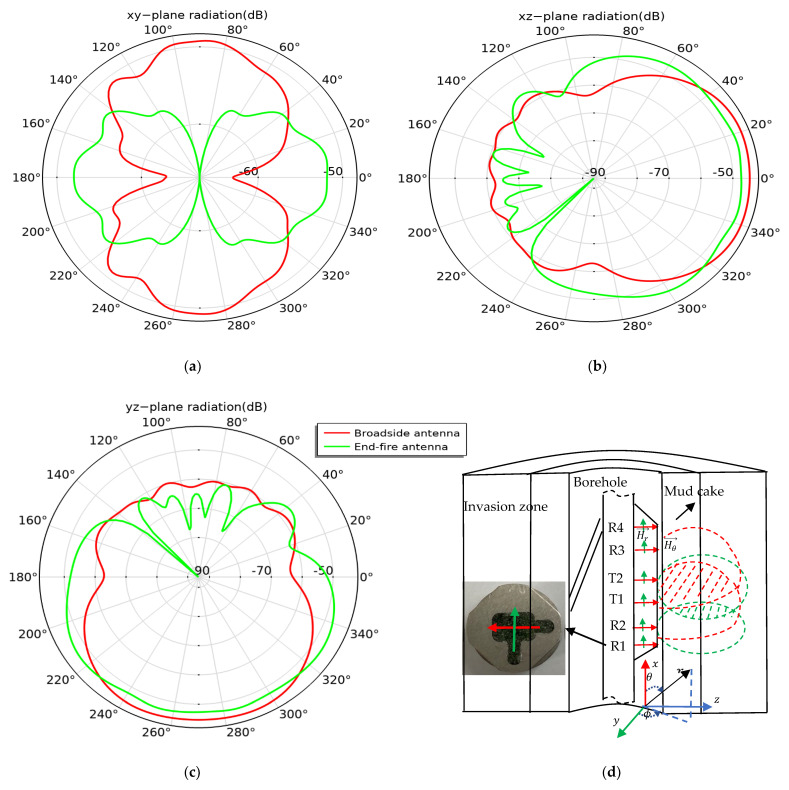
Two-dimensional radiation plot and detection depth of the antenna: (**a**) *xy*-plane radiation; (**b**) *xz*-plane radiation; (**c**) *yz*-plane radiation; (**d**) schematic diagram of detection depth of antenna.

**Figure 5 sensors-22-07667-f005:**
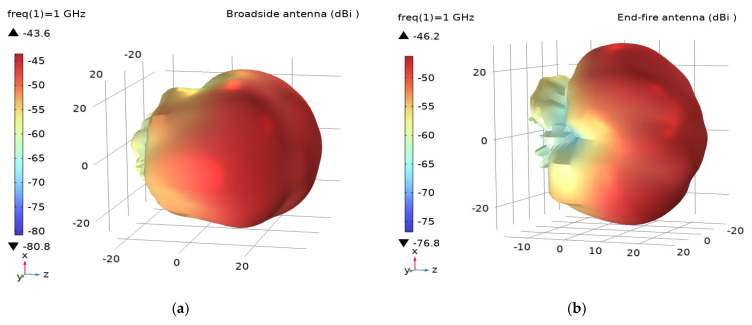
Three-dimensional radiation patterns of antenna: (**a**) 3D radiation patterns of the broadside antenna; (**b**) 3D radiation patterns of the end-fire antenna.

**Figure 6 sensors-22-07667-f006:**
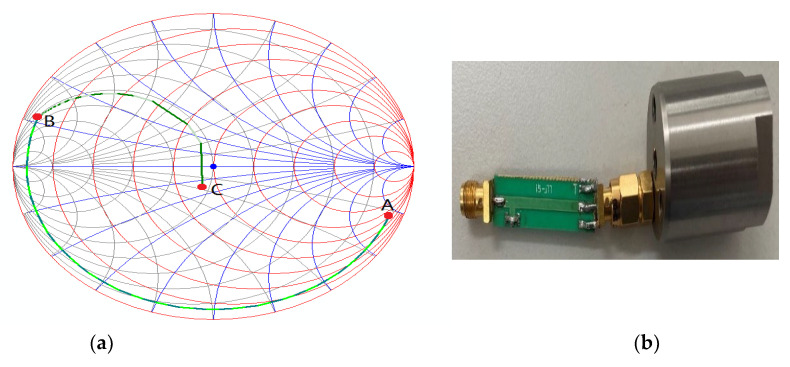
Antenna impedance matching and physical image: (**a**) impedance matching process using a microstrip line and a parallel capacitor; (**b**) physical image of the antenna and matching circuit.

**Figure 7 sensors-22-07667-f007:**
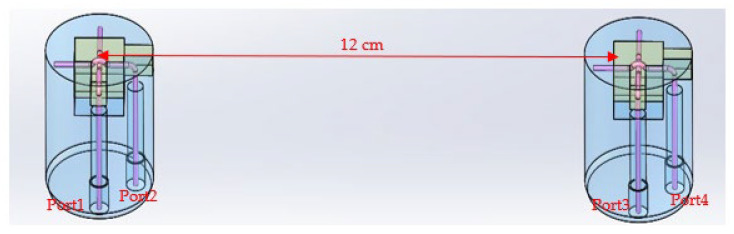
Simulation setup for dual antennas spaced 12 cm apart in water.

**Figure 8 sensors-22-07667-f008:**
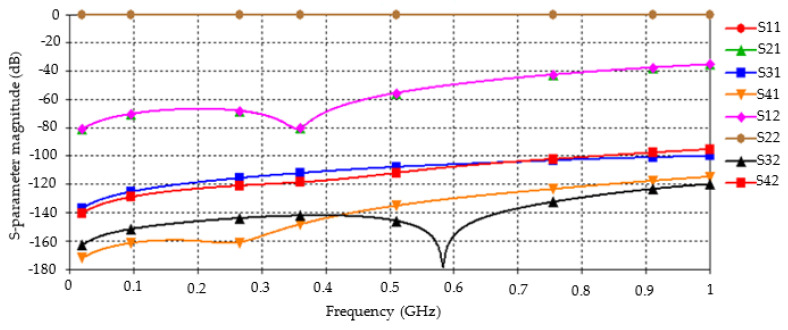
Simulation curve of transmission parameters of dual antennas.

**Figure 9 sensors-22-07667-f009:**
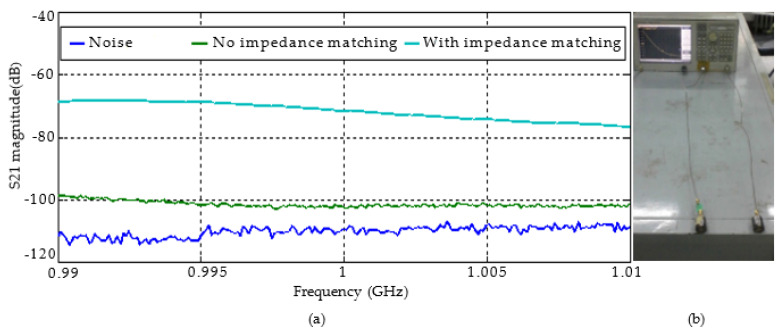
Actual measurement of transmission parameters of dual antennas: (**a**) measured curve of transmission parameters of dual antennas; (**b**) actual measurement diagram.

**Figure 10 sensors-22-07667-f010:**
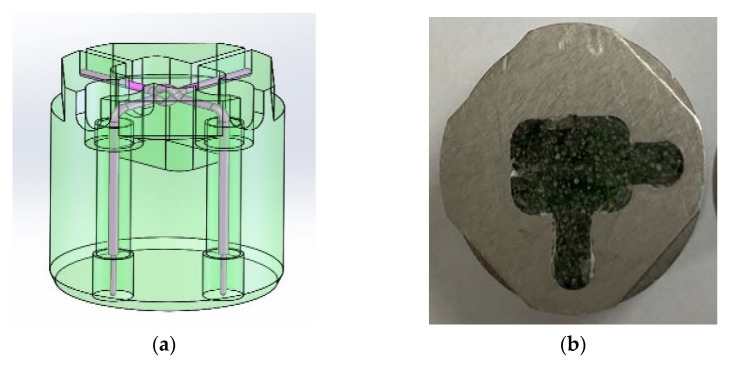
Antenna design and development image: (**a**) 3D design drawing; (**b**) actual development image diagram of antenna.

**Figure 11 sensors-22-07667-f011:**
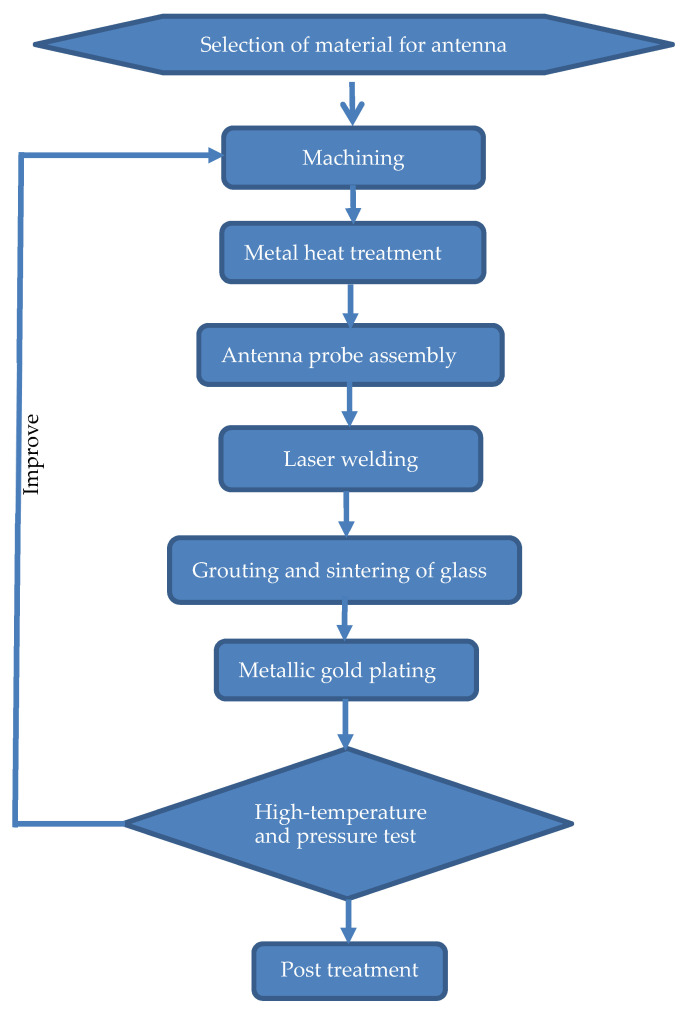
Workflow for development of a suitable antenna.

**Figure 12 sensors-22-07667-f012:**
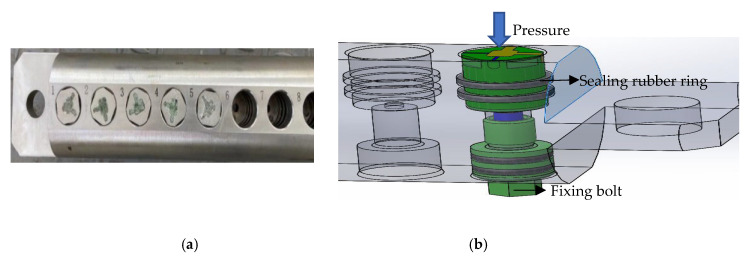
Antenna performance verification device: (**a**) the verification device; (**b**) 3D model of verification device.

**Figure 13 sensors-22-07667-f013:**
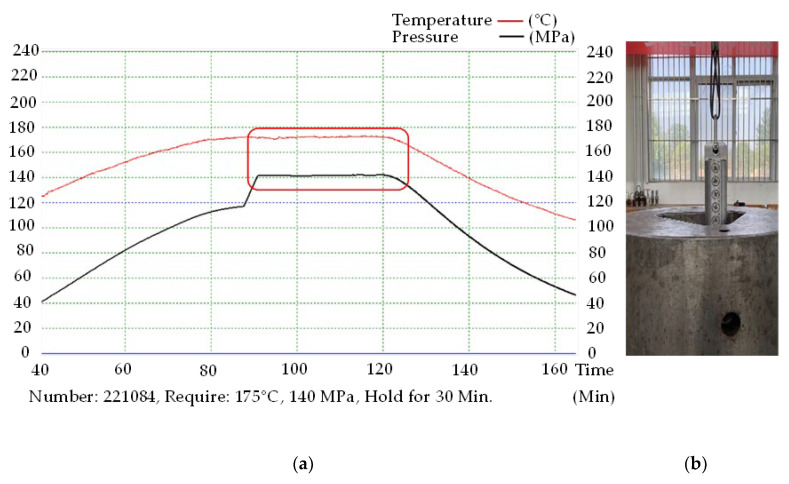
High-temperature and pressure resistance test: (**a**) high-temperature and pressure resistance test curve of antenna; (**b**) actual measurement diagram.

**Figure 14 sensors-22-07667-f014:**
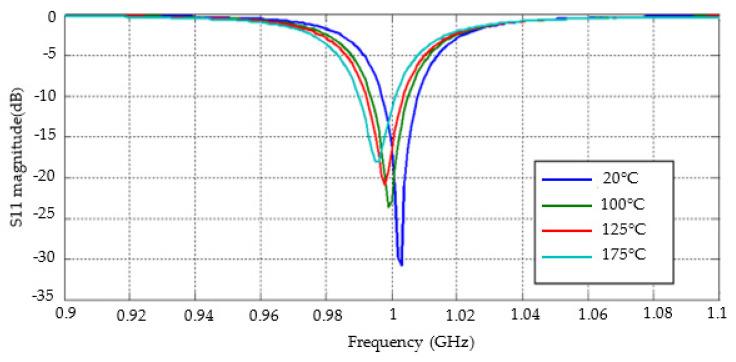
The S11 amplitude of the developed antenna varies with temperature.

**Figure 15 sensors-22-07667-f015:**
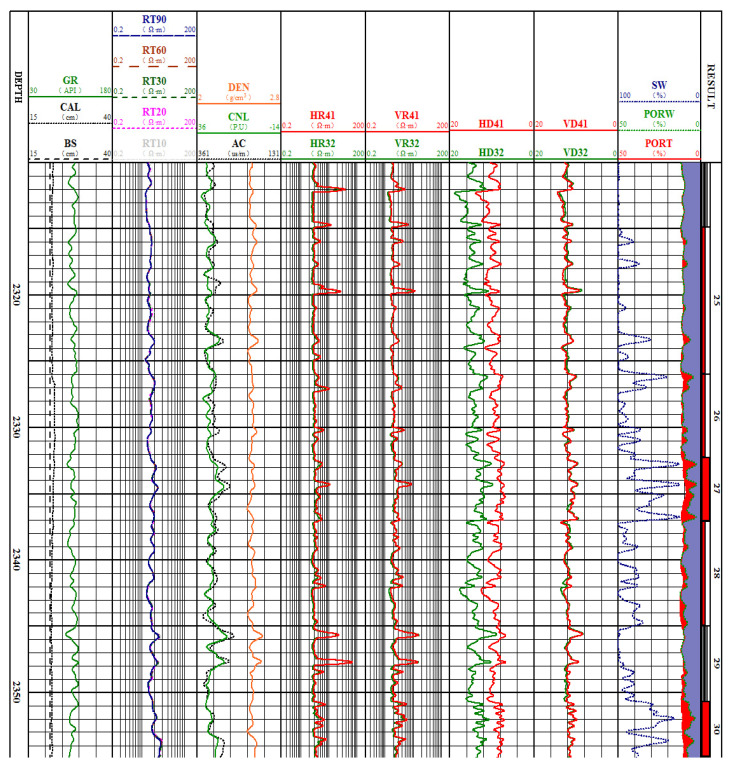
The dielectric logging sensor using the proposed antenna applied in the oilfield.

## Data Availability

Not applicable.

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
