# Peer review of "Development and Application of Dual-Polarization Antenna for Dielectric Logging Sensor"

_sensors, 2022, doi:10.3390/s22197667_

Round 1
Reviewer 1 Report
The manuscript should be reorganized and well written. It hinders the understanding because of the improper content arrangements as well as English expression.
The main contribution of this study is to propose a dual polarization antenna for dielectric logging sensor. But the manuscript does not give the design idea of the antenna structure, does not expatiate the different characteristic of the antenna compared to the published works, and does not explain why the proposed sensor can adapt to the application environment with high temperature and high pressure. The scientific paper is different from that of the report, so the highlight should be summed up carefully.
In addition to those critical issues, here are further comments:
1. What functions are these modules respectively represented by the different colors and shapes in Figure 2? What materials are used for each module? Which structures or dimensions will have a more significant impact on the antenna performances?
2. How to understand the flow path of the current shown in green in FIG. 3(a)?
3. Eq. (1) ~ (3) are general expressions of antenna radiation. It is hoped to give the particular calculation formula for the proposed antenna structure.
4. What is the relationship between the structures shown in Fig. 4d and in Fig. 2?
5. There are no comparative analysis in the manuscript, but some conclusions were drawn suddenly. For example
1) The above-mentioned indicates that the detection depth of the end-fire antenna is greater than that of the broadside antenna.
2) … shown in Figure 5, which indicates that the antenna with the working frequency 1GHz will have the strongest electric field radiation along the probe direction with good directionality …
Author Response
Dear Reviewer:
Please see the attachment

Reviewer 2 Report
There is sufficient work reported in the manuscript. However, the reported antenna and the way of implementation is inappropriate such that one cannot reimplement it. Please elaborate the design guidelines, dimensions and structure in clearer manner.
Author Response
Dear Reviewer:
Please see the attachment.

Reviewer 3 Report
In the manuscript entitled “Development and application of dual polarization antenna for dielectric logging sensor”, the authors designed and developed a dielectric logging sensor based on the physical properties of a dual polarized slot antenna. The authors thoroughly described the antenna design, their physical properties, radiated beam and working principles. Moreover, the authors developed and tested their model under high-pressure and temperature conditions, demonstrating the feasibility of their concept. The contribution of this work, in relation to the available literature, was properly contextualized in the introduction section. The abstract and conclusions sections agree with the content in the main text. From my point of view, this work is an interesting contribution to the area of logging-sensors. Therefore, I recommend publication after the following minor changes:
· The authors mention red and green arrows in Figure 1. However, at first glance only red and blue arrows appear in this figure. It was only after zooming in a lot in Figure 1 that I was able to notice the very small green arrows. Therefore, authors should review the presentation of this figure to improve readability.
· The authors used physical units (eg, m or mm) in italics, which is inappropriate as physical units are not mathematical parameters.
Author Response

(The authors gave the same response as above.)

Round 2
Reviewer 1 Report
The quality of the revised manuscript has been improved, but several core issues mentioned in the last review were not addressed.
1. In the response letter, the author expounds the improvement of material selection and manufacturing processes of the sensor. But the reader still has no idea about the design philosophy of the dual polarization antenna, and the improvement of the proposed structure relative to the existing design.
2. The names and materials of the parts in different colors in Figure 2 are added in the revised manuscript. In fact, the reader is more concerned about why to design the topological structure shown in Figure 2 and how to choose the corresponding size for each part.
Author Response

(The authors gave the same response as above.)

Reviewer 2 Report
all the points have been incoporated
Author Response

(The authors gave the same response as above.)
